Patient satisfaction after outpatient hysteroscopy: a retrospective descriptive study

Sanchez Carbonell Claudia
Rovira Pampalona Jennifer jrovira@csa.cat
Oliveres Amor Carla
Caballol Arteaga Alexandra
Degollada Maria
Brescó Torras Pere
1 Obstetrics and Gynecology Department, Hospital Universitari d’Igualada , Igualada , Spain
2 Fundació Sanitaria d’Igualada, Fundació Sanitaria d’Igualada , Igualada , Barcelona , Spain
Anson Lesley
Electronic publication date: 2025 Nov 10
Publication date: 2025
Volume: 13
Electronic Location ID: e20272
Received 2025 Mar 26; Accepted 2025 Sep 30
Copyright: ©2025 Sanchez Carbonell et al.
Copyright year: 2025
Copyright holder: Sanchez Carbonell et al.
License: This is an open access article distributed under the terms of the Creative Commons Attribution License, which permits unrestricted use, distribution, reproduction and adaptation in any medium and for any purpose provided that it is properly attributed. For attribution, the original author(s), title, publication source (PeerJ) and either DOI or URL of the article must be cited.
License URL: https://creativecommons.org/licenses/by/4.0/

Keywords: Patient satisfaction, Office hysteroscopy, Hysteroscopy

Funding: The authors received no funding for this work.

==============================
Background

The aim of the study is to analyse the overall satisfaction level of patients undergoing diagnostic and/or therapeutic hysteroscopy in an ambulatory setting and examine factors related to satisfaction.

Methods

A retrospective descriptive study was conducted to analyse outpatient hysteroscopies performed between January 2020 and June 2022 at the University Hospital of Igualada. Patient demographic and clinical data as well as hysteroscopic features were collected. A telephonic questionnaire on patient satisfaction was conducted retrospectively.

Results

A total of 435 hysteroscopies were analysed. Hysteroscopy was successful in 95.6% of them with a clinical remission in 69.8% of patients. The mean pain score was 3.33 (Visual Analogue Scale). An average overall satisfaction score of 9 was obtained. Pain was the main reason in patients with low satisfaction ratings. A positive correlation was found between the patient satisfaction score and the level of information received before the procedure. An inverse relationship was detected between the patient satisfaction score and the pain experienced during the hysteroscopy.

Conclusions

Outpatient diagnostic and/or therapeutic hysteroscopy is a technique accepted by the majority of patients and with a high level of satisfaction. Variables such as pain or the previous information received are important and directly related to the final satisfaction level of the procedure.

Background

Hysteroscopy is a highly useful technique for the diagnosis and treatment of intrauterine pathology, considered the gold standard for endometrial pathology (Vitale et al., 2021).

Over the last decade, there has been a substantial increase in interest in performing most procedures in an ambulatory setting (Vitale et al., 2020a). Technological advances with smaller-calibre hysteroscopes and various instruments as therapeutic tools have significantly enabled this paradigm shift.

Outpatient hysteroscopy is an efficient procedure with advantages over the same procedure performed in a conventional operating room. These advantages include the absence of hospitalization and preoperative tests, and also removing any risks associated with regional or general anaesthesia. This allows for a quicker post-surgical recovery with an overall lower procedure cost (Marsh, Rogerson & Duffy, 2006; Smits et al., 2016; Yang & Chaudhari, 2020; Shields et al., 2018; Bennett et al., 2019; De Silva et al., 2020). Notably, there is also a lower rate of complications such as cervical tears, uterine perforations and complications associated with distension media (Cholkeri-Singh & Sasaki, 2016; Shields et al., 2018; Raz et al., 2022), making it a compelling reason to recommend outpatient hysteroscopy whenever possible.

Several studies have demonstrated its safety and an equivalent success rate to hysteroscopy in the operating room. Although most reviews refer to outpatient hysteroscopy as a well-tolerated and accepted procedure, the fact that it can be an invasive and painful procedure for some patients should not be undervalued. Patients may experience pain during manipulation or entry through the internal cervical orifice, cavity distension, and the hysteroscopic procedure itself (Guida et al., 2006; Smith et al., 2019).

The acceptable limit of pain in an ambulatory setting and its impact on patient satisfaction raise important questions. There is a limited body of literature focusing on patient satisfaction with diagnostic and/or therapeutic outpatient hysteroscopy (De Silva et al., 2021). Hence, there is a gap in the literature to assess the appropriateness of the procedure, identify potential improvements, and support the idea that most procedures can be performed in an office setting (Nanayakkara et al., 2022).

The aim of our study is to analyse the overall satisfaction level of our patients undergoing diagnostic and/or therapeutic hysteroscopy in an ambulatory setting, and to examine which factors could be related to patient satisfaction.

Materials and Methods

Study design

We conducted a retrospective descriptive study analysing outpatient hysteroscopies performed between January 2020 and June 2022 at the University Hospital of Igualada.

All patients who underwent outpatient hysteroscopy during the study period, regardless of the indication, were contacted by phone and asked to participate in the study that included data review and the completion of a telephone-administered satisfaction questionnaire. Patient consent was obtained verbally after explaining the study. Patients who did not answer the call or expressed unwillingness to participate were excluded.

Hysteroscopy procedure

All hysteroscopies were performed in office, without local anaesthesia or cervical preparation, as per the hospital protocol, as well as applying vaginoscopy. Physiological normal saline (0.9%) was used as distending media, using gravity flow system.

The hysteroscopes used included the Truclear System 5C Mechanical Energy Hysteroscope (5.7 mm), Olympus Hysteroscope (5.3 mm) with forceps, scissors or the Versapoint electrosurgery system, Delmont (4–5.5 mm) and Endosee (4.3 mm), chosen according to the preference of the performing physician. The responsible physician for the procedure was a specialist or a fourth-year resident rotating through the service during the study period.

Successful hysteroscopy was defined as the procedure that could be completed in office, without the need to reprogram an additional hysteroscopy in the operating room. VAS pain scores were assessed at the time of cavity entry, during the procedure, and 10 min after finishing the procedure.

Data collection

Patient and hysteroscopy-related data were collected from computerized medical records and a computerized questionnaire linked to the patient’s medical record that is always completed at the end of every hysteroscopy.

Demographic and clinical data were collected, including age (years), parity, history of cesarean section (yes/no), hormonal status (premenopausal/menopause), clinical presentation (asymptomatic, abnormal uterine bleeding, infertility, retained IUD, retained products of conception), and a history of anxiety or depression (yes/no).

Data related to hysteroscopy were also analysed, including the type of hysteroscope used, performing physician, indication for hysteroscopy (pathological and/or clinical ultrasound, infertility/uterine malformation/isthmocele study, pathological biopsy/hyperplasia control/abnormal cervical cytology, retained IUD), procedure performed (diagnostic, polypectomy, myomectomy, removal of retained products of conception, IUD extraction, septoplasty), successful hysteroscopy (yes/no), and patient-perceived pain (visual analogue scale, VAS, 0–10).

The telephone survey was conducted retrospectively between August and December 2023. The survey assessed the information received before the procedure (sufficient, intermediate, insufficient, does not remember), whether the patient would undergo the procedure in the office again (yes/no), whether she would recommend the procedure (yes/no), clinical remission (total, partial, no remission, worsening), and the overall satisfaction level (0–10).

Statistical analysis

All variables were recorded in an anonymized Excel database and analysed using Jamovi (version 2.3.21.0). Absolute and relative frequency for each categorical variable were calculated, while median and interquartile range were used for continuous variables. Patient satisfaction was compared across different variables using the Kruskal–Wallis test; in those with a significant relationship, a Spearman correlation test was performed. Additionally, a multivariable analysis (multiple linear regression) was performed using SPSS (version 19.0; IBM Corporation, Chicago Illinois).

Ethical statement

The study adhered to the ethical and research recommendations outlined by the Declaration of Helsinki. Measures were taken to ensure the privacy of patient identifiable data and collected information. Data processing was carried out in accordance with the Organic Law 3/2018 Protection of Personal Data and approved by the Data Protection Officer of the hospital. The study received approval from the corresponding Clinical Research Committee of the University Hospital of Igualada and the Ethics Committee of the University Hospital of Bellvitge (AC147/12). Patient consent was obtained verbally.

Results

A total of 720 hysteroscopies were performed during the study period. Patients who did not answer the phone call (278) and those who declined to participate in the satisfaction survey (seven) were excluded from the study. Consequently, 435 hysteroscopies were analysed (Fig. 1).

Figure 1 CONSORT diagram.

Patient recruitment flow chart.

Retrospective analysis of hysteroscopy procedures

Patient characteristics

The mean age of the patients was 49 years, with 56.8% of them being premenopausal (Table 1). The main reason for hysteroscopy was the symptomatology and/or pathological ultrasound findings (85.0%).

Table 1 Patient characteristics.

Patient characteristics	Median (IQR)/N (%)	
Age	49 (42.0–57.0)	
Menopausal state		
Pre-menopause	247 (56.8%)	
Menopause	188 (43.2%)	
Nº vaginal deliveries		
0	98 (22.5%)	
1	81 (18.6%)	
≥ 2	256 (58.9%)	
Cesarean section		
Yes	86 (19.8%)	
No	349 (80.2%)	
Anxiety/Depression		
Yes	98 (22.5%)	
No	337 (77.5%)	

Procedure characteristics

Hysteroscopy was successful in 95.6% of patients; only 10 patients were referred for surgical hysteroscopy with anaesthesia. The Truclear System was the most commonly used hysteroscope in the majority of procedures (51.2%). The most frequent finding was endometrial polyps (48.0%) (Table 2).

Table 2 Procedure characteristics.

Procedure characteristics	Median (IQR)/N (%)	
Hysteroscopist		
Attending	332 (76.3%)	
Resident	103 (23.7%)	
Hysteroscope		
Truclear System	223 (51.2%)	
Olympus +/- Versapoint	140 (32.2%)	
Delmont	56 (12.9%)	
Endosee	16 (3.7%)	
Surgical procedure		
Polypectomy	209 (48.0%)	
Biopsy/no procedure	183 (42.1%)	
Myomectomy (partial/total)	18 (4.1%)	
Retained products of conception	14 (3.2%)	
IUD Extraction	10 (2.3%)	
Septoplasty	1 (0.2%)	
Successful procedure		
Yes	416 (95.6%)	
No	19 (4.4%)	
Pain score during hysteroscopy	
VAS entry	4.0 (2.0–7.0)	
VAS during procedure	3.0 (2.0–6.0)	
VAS after	2.0 (1.0–4.0)	
Median VAS	3.33 (2.0–5.0)	
Notes.

IUD Intauterine device

VAS Visual Analogue Scale

The evaluation of pain resulted in a mean score of 3.33 on the VAS scale. The highest score was reported during cavity entry (VAS 4), followed by a pain score of 3 on the VAS scale during the procedure, and a VAS 2 once the intervention was completed (Table 2).

Patient satisfaction analysis

The post hoc telephone satisfaction survey featured an average overall satisfaction score of 9. Patients reporting low satisfaction (score ≤5, 11.5%), cited pain as the main reason (78.0%). When asked if they would undergo the procedure again, if need be, 74.0% of patients answered affirmatively. Additionally, 80.2% would recommend the procedure. Clinical remission was complete in 71.0% of patients. When asked about the information received regarding the procedure and intervention, only 65.3% reported having sufficient information before the procedure (Table 3).

Table 3 Satisfaction survey results.

Patient satisfaction	N (%)/Median (IQR)	
Previous information		
Sufficient	284 (65.3%)	
Intermediate	79 (18.2%)	
Insufficient	67 (15.4%)	
Does not remember	5 (1.1%)	
Would repeat in office procedure	
Yes	322 (74.0%)	
No	93 (21.4%)	
Doesn’t know	20 (4.6%)	
Would recommend procedure	
Yes	349 (80.2%)	
No	86 (19.8%)	
Clinical remission.N = 286(149 not applicable)	
Total	203 (71.0%)	
Partial	50 (17.5%)	
No remission	30 (10.5%)	
Worsening	3 (1.0%)	
Satisfaction level	
Median Satisfaction	9.0 (8.0-10.0)	
Low (0–5)	50 (11.5%)	
Good (6–8)	155 (35.6%)	
Excellent (9–10)	230 (52.9%)	
Reason for low satisfaction (≤5).N = 50	
Pain	39 (78.0%)	
Lack of information	4 (8.0%)	
Personal treatment	1 (2.0%)	
Other	6 (12.0%)	

The analysed data showed no statistically significant correlation when comparing the satisfaction level to a history of depression or anxiety (7.72 vs. 8.25; p = 0.126) nor depending on menopausal status (8.01 vs. 8.26; p = 0.427). No statistically significant differences were found in the patients’ satisfaction level related to the professional performing the procedure (attending 8.2 vs. resident 7.99; p = 0.188), nor depending on the performed procedure (p = 0.173).

Contrarily, significant differences were found in the correlation between the patient satisfaction score and the level of information received before the procedure, with a positive relationship (ρ Spearman −0.329, P <.001; Fig. 2). Not surprisingly, an inverse relationship was detected between the patient satisfaction score and the pain experienced during the hysteroscopy (ρ Spearman −0.238, P <.001; Fig. 3).

Figure 2 Correlation between “Previous information” and “Satisfaction” (Spearman correlation).

ρ Spearman −0.329, p < 0.001. Previous information (sufficient, intermediate, insufficient, doesn’t remember). Patient satisfaction (score 0–10).

Figure 3 Correlation between “Pain” and “Satisfaction” (Spearman correlation).

ρ Spearman −0.238, p < 0.001. Patient satisfaction expressed in a scale 0–10. Pain expressed as median VAS (Visual Analogue Scale, score 0–10).

In the multivariable analysis regarding the satisfaction grade, we found no significant relationships between clinically relevant variables (Table S1).

Discussion

The present study has found a great level of satisfaction in office hysteroscopy (overall satisfaction of 9/10). It also demonstrates the importance of previous information given to the patient and the influence of pain during the procedure as two factors affecting final satisfaction.

Over the last decades, surgical innovations in hysteroscopy have considerably changed the approach to intrauterine pathology. In the past, with a “see and treat” approach, only minor pathologies were treated in an ambulatory setting, leaving more complex procedures for the operating room. However, with the advent of new outpatient operating technology—such as new hysteroscopes, mini-resectoscopes, uterine morcellators, diode lasers, miniaturized mechanical instruments, and endometrial ablation devices—office hysteroscopy has been accepted as a feasible and effective tool for treating almost all intrauterine pathologies (Vitale et al., 2020a).

In our study, different hysteroscopic techniques were used, the most common being the hysteroscopic morcellator for endometrial pathology, followed by mechanical instruments such as forceps and scissors. The ability to choose the technique based on the pathology to be treated, patient characteristics, and the experience of the responsible physician can facilitate the procedure. In our study, this is reflected in the high success rate of outpatient hysteroscopies (95.6%), in line with previous literature (94–98% success rate (Bougie et al., 2013; Moawad et al., 2014; Sofoudis et al., 2023; Wright, Hamilton & Kosturakis, 2024))- and the overall satisfaction of patients (9/10), as reported in other studies (Pervaiz et al., 2021; Wright, Hamilton & Kosturakis, 2024). Another variable that could be associated with the high degree of satisfaction could be the elevated rate of symptom resolution (88.5% of patients in our study had their symptoms improved after the hysteroscopy).

Another factor for the high satisfaction could be the fast resolution of the process through the “see and treat” approach, in addition to the short waiting list for office hysteroscopy in our centre.

In the literature there are heterogenous results on patient satisfaction, however, all studies report a high satisfaction rate in outpatient hysteroscopy (4.9/5 (Bougie et al., 2013); 89.7% of high satisfaction (Filiz et al., 2009)); 95.5% of acceptability (Bougie et al., 2013; Pervaiz et al., 2021). In our study, 74% of patients would repeat the procedure in office and 80% would recommend it. Similar conclusions are reported in the literature (Kremer, Duffy & Moroney, 2000; Wortman & Daggett, 2012; Pervaiz et al., 2021).

This study was prompted by the absence of consensus regarding the acceptable pain threshold for outpatient procedures. We know that hysteroscopy is an invasive test that can be painful and perceived as a negative experience by some patients, being this the main reason for patient dissatisfaction. In our study, pain was the main reason for dissatisfaction or not wanting to go through the same procedure if need be. The average pain score in our study was 3.33 on the VAS scale. Although some studies report that the level of pain experienced doesn’t have an impact on patient satisfaction (Akca et al., 2020; Pervaiz et al., 2021), our findings point in the opposite direction, as we found a negative correlation between the two variables.

Multiple variables are considered risk factors for pain, including clinical and anatomical characteristics such as parity, menopausal status, dysmenorrhea, chronic pelvic pain, endometriosis, as well as non-clinical characteristics such as emotional state, anxiety, patient preferences, and past experiences. All of these are variables directly related to the perception of pain and patient satisfaction (Gambadauro, Navaratnarajah & Carli, 2015; Vitale et al., 2020b; Akca et al., 2020; Sorrentino et al., 2021). Of these, preoperative anxiety has been the focus of many studies, being reported as a risk factor for increased pain perception (Gambadauro, Navaratnarajah & Carli, 2015; Akca et al., 2020). Moreover, previous information can reduce preoperative anxiety and thus, improve patient satisfaction. In our study, we didn’t collect information of anxiety level prior to the procedure, but we found no difference in satisfaction considering history of anxiety/depression.

In our study, an association was found between patient satisfaction and previous information given. Patients with low satisfaction ratings referred insufficient information previous to the outpatient hysteroscopy. Positive correlation between preoperative explanation and patient satisfaction was also found in other studies (Pervaiz et al., 2021). Hence, the significance of pre-procedure counselling should not be underestimated. Patients should be informed about potential discomfort related to the procedure and the possibility of interrupting it if not well tolerated (with the option to reschedule for a subsequent hysteroscopy in the operating room with sedation), as previous knowledge could positively influence the pain experienced (Pervaiz et al., 2021). This is particularly important in the era of social media, where patients may look up information online. Several studies report that a great percentage of social media content is not evidence-based and could lead to misinformation (Adler et al., 2024; Vitale et al., 2025a; Vitale et al., 2025b). This may have an influence on fear, anxiety and unrealistic expectations for patients, which can have a negative impact on patient satisfaction, especially in outpatient procedures. It is important to raise awareness about the low quality of information in social media, to increase health-professional’s reliability and trustworthiness.

This study has several notable features. To our knowledge, it is one of the few studies that evaluates the overall and specific satisfaction levels of different hysteroscopic procedures in the office without any anaesthesia or cervical preparation. Additionally, the high overall satisfaction score and successful rate obtained confirm the well-known advantages of outpatient hysteroscopy, emphasizing patient preferences and experience in selecting treatment options and avoiding the risks and inconveniences associated with the surgical environment.

There is currently no standardized guide or protocol on selection criteria for outpatient hysteroscopy (on which patients or pathologies) or when it is preferable to perform it in the operating room; it depends on each centre and the experience of the responsible physician. This study emphasizes the importance of good pre-procedure information and the possible incorporation of analgesic/anaesthetic techniques based on the patient and pathology to be treated.

We consider relevant the fact that patient satisfaction was evaluated both quantitative and qualitative: not only with a general punctuation (0–10), but also through specific questions, like if they would recommend the procedure or would undergo it again.

Finally, as a strength of this study, we would point out that all patients who were selected for an outpatient hysteroscopy were included, independently of the indication; therefore, it is a representative sample of the daily practice.

As a limitation, we would emphasize that the questionnaire was conducted via telephone and retrospectively, which may hinder recruitment and introduce recall biases, respectively. There could be a selection bias due to the telephone survey, as 285 out of 720 hysteroscopies weren’t evaluated because the patient didn’t answer the phone or declined to participate. Another bias could be the time passed since the procedure, as it was not the same for every woman (some patients had had the hysteroscopy 1 year before the survey while others, 3 years before); the amount of time passed could induce memory bias and alter the sensations perceived a long time ago.

Conclusion

Outpatient diagnostic and/or therapeutic hysteroscopy is a technique accepted by the majority of patients with a high level of satisfaction. Variables such as perceived pain or the previous information received are important and directly related to the final satisfaction level of the procedure.

Supplemental Information

Supplemental Information 1 Satisfaction questionnaire (Spanish and English translation)

Supplemental Information 2 Multiple linear regression table

Additional Information and Declarations

Competing Interests

Author Contributions

Human Ethics

Data Availability

The authors declare there are no competing interests.

Claudia Sanchez Carbonell conceived and designed the experiments, performed the experiments, analyzed the data, prepared figures and/or tables, authored or reviewed drafts of the article, and approved the final draft.

Jennifer Rovira Pampalona conceived and designed the experiments, performed the experiments, analyzed the data, prepared figures and/or tables, authored or reviewed drafts of the article, and approved the final draft.

Carla Oliveres Amor performed the experiments, analyzed the data, prepared figures and/or tables, authored or reviewed drafts of the article, and approved the final draft.

Alexandra Caballol Arteaga performed the experiments, authored or reviewed drafts of the article, and approved the final draft.

Maria Degollada performed the experiments, authored or reviewed drafts of the article, and approved the final draft.

Pere Brescó Torras performed the experiments, authored or reviewed drafts of the article, and approved the final draft.

The following information was supplied relating to ethical approvals (i.e., approving body and any reference numbers):

The Bellvitge University Hospital Clinical Research Ethics Committee approved to carry out the study (Ref: AC147/12)

The following information was supplied regarding data availability:

The raw data is available at Zenodo and OSF:

– Sanchez Carbonell, C. (2025). Hysteroscopy patient satisfaction Database [Data set]. Zenodo. https://doi.org/10.5281/zenodo.16739175

– https://osf.io/ejhv6/overview

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
