# Peer review of "Patient satisfaction after outpatient hysteroscopy: a retrospective descriptive study"

_PeerJ, doi:10.7717/peerj.20272_

## Round 0.1 · original submission · Major Revisions

· Academic Editor

Major Revisions

·

Basic reporting

The article is written in English and uses clear, technically correct text.
The article includes relevant work, but in some sections there's lack of comparison of results with prior similar studies regardind this topic
- Pervaiz Z, Korrapati S, Ghoubara A, Ewies A. Office hysteroscopic morcellation service: Evaluation of women experience and factors affecting satisfaction. Eur J Obstet Gynecol Reprod Biol. 2021 Sep;264:294-298. doi: 10.1016/j.ejogrb.2021.07.049. Epub 2021 Jul 28. PMID: 34352426.
- Akca A, Yilmaz G, Esmer AC, Yuksel S, Koroglu N, Cetin BA. Use of video-based multimedia information to reduce anxiety before office hysteroscopy. Wideochir Inne Tech Maloinwazyjne. 2020 Jun;15(2):329-336. doi: 10.5114/wiitm.2019.89378. Epub 2019 Oct 28. PMID: 32489494; PMCID: PMC7233155.
- Gambadauro P, Navaratnarajah R, Carli V. Anxiety at outpatient hysteroscopy. Gynecol Surg. 2015;12(3):189-196. doi: 10.1007/s10397-015-0895-3. Epub 2015 May 13. PMID: 26283891; PMCID: PMC4532701.
The article's structure conforms to an ecceptable format and include all results relevant to the hypothesis

Experimental design

The research question is well defined and the investigation has been conducted with high standards.
Methods are well described.

Validity of the findings

No comments

Additional comments

Here are my reccomendations regarding the possible improvements in the manuscript

Abstract.
Line 31-33 - The results section on the abstract needs to be reorganized as it will be more structured. I suggest : 435 hysteroscopies were analysed. Hysteroscopy was successful in 95.6% of them with a clinical remission in 69.8% of patients. The
mean pain score was 3.33 (Visual Analogue Scale). An
average overall satisfaction score of 9 was obtained (…)

Materials & Methods - Study design
Line 89—93 The explanation of the study design is a llitle bit confusing. we suggest rephrasing the paragraph to make it more understandable. as a suggestion wee propose : “ All patients who underwent outpatient hysteroscopy during the study period, regardless of the indication, were contacted by phone and asked to participate in the study that included data review and the completion of a telephone-administered satisfaction questionnaire. patient consent was obtained verbally
after explaining the study. Patients who did not answer the call or expressed unwillingness to participate were excluded.

Materials & Methods - Data Collection
Line 99 - The acronym TPAL is not explained before in teh manuscript, we suggest to evplain it or remove , as it is not often used all over the world.
Line 99 - The term “C-section” should be changed to “hystory of cesarean section”
Line 102 - te words “were collected” sholud go on line 99 “Demographic and clinical data werw collected (…)”
Lines 110-120 - Those lines refer to how the hysteroscopic procedures were performed, not regardind the data collection. We suggest to elaborate a separate pragraph (Hysteroscopy procedure) explaining how the procedures are performed the center (anestesia, location, fluid distension media, pressures, type of hysteroscope and their diameters, performers of the procedure,…) That paragraph should go before the data collection part.
Lines 122-126 - those lines refer to the data collection, so we reccomend to be moved after line 109.

Results
Figure 1 - This representation is called a CONSORT diagram. It should be added to the explaining text below. On the diagram, there’s some issue regarding the percentages, as they together equal more than 100%. They should be rcaclculated or better explained

We propose changing the structure of the Results part to something like that:
Results
Retrospective analisis of hysteroscopy procedures
Content from lines 149-157
Patient satisfaction analysis
Content from lines 158-173

Line 149-157 - All data explained in that lines are from the hysteroscopic procedures but not related with the aim of the study (“analyse the overall satisfaction level of our patients undergoing diagnostic and/or therapeutic hysteroscopy in an ambulatory setting, and to examine which factors could be related to patient satisfaction”). We propose to refer to that data as “Patient characteristics” and “Procedure characteristics” and put in a separate part.
Line 158-173 - Those data refer to the aim of the study , so it should be put in an independent part.

Line 165-168 - The afirmations regarding correlations stated in this sentence are not shown in any table or supplementary material. If the authors want to include the afirmations, we reccomendo to write the results and significance or including a table/figure with all the results.

Discussion
The Discussion part of a manuscript should begin always with a statement regarding the results observed in the present study ( p.e. “The present study has found….”). we recommend to include that and after that begin with the line 177.

Line 181 - The sentence “new outpatient operating technology- such as hysteroscopes, mini-resectoscopes (…)” should include the word “new” (or similar )before “hysteroscopes”.

Lines 188-189 - This sentence should indivctae that this resultas are referred to the center of the study. It will be also good to make some reference to other studies with similar results

Line 189-191 - This sentence is difficult to understand. we suggest some clarification for a better understanding

Line 192-193 - This sentence is referring to a see-and-treat standard of care? Maybe should be expressed as : Another factor for the high satisfaction could be the fast resolution of the process through the ‘see and treat’ approach
Line 194-195 - The data between parethesiss should be at the end of the sentence : “… satifaction rate in outpatient hysteroscopy ((4.9/5 [15]; 89.7% of high satisfaction [16]) and
195 procedure success (>96% [15])”

Line 198-199 - A clarification that these are data form your study should be added. In addition, these sentence make more sense if tehy are added after the paragraph 200-207.

Lines 208-213 - The discussion section should includ some references to previous knowlegde and other studies adressing the same issues.
We suggest to add some references to the results of theses studies and make some comments about having or not similar results:
Pervaiz Z, Korrapati S, Ghoubara A, Ewies A. Office hysteroscopic morcellation service: Evaluation of women experience and factors affecting satisfaction. Eur J Obstet Gynecol Reprod Biol. 2021 Sep;264:294-298. doi: 10.1016/j.ejogrb.2021.07.049. Epub 2021 Jul 28. PMID: 34352426.
Akca A, Yilmaz G, Esmer AC, Yuksel S, Koroglu N, Cetin BA. Use of video-based multimedia information to reduce anxiety before office hysteroscopy. Wideochir Inne Tech Maloinwazyjne. 2020 Jun;15(2):329-336. doi: 10.5114/wiitm.2019.89378. Epub 2019 Oct 28. PMID: 32489494; PMCID: PMC7233155.
Gambadauro P, Navaratnarajah R, Carli V. Anxiety at outpatient hysteroscopy. Gynecol Surg. 2015;12(3):189-196. doi: 10.1007/s10397-015-0895-3. Epub 2015 May 13. PMID: 26283891; PMCID: PMC4532701.

Line 231-232 - Maybe you should consider another bias : time passed from the procedure, as the study was conducted between 1 and 3 years after the procedure was perfromed. That amount of time could have some bias on the sensations perceived long time ago.

Reviewer 2 ·

Basic reporting

The manuscript is written in clear and professional English. The structure adheres to PeerJ guidelines, and the introduction provides sufficient background with relevant and recent literature references. Figures and tables are appropriate, well-labelled, and support the narrative effectively.

However, the title should explicitly reflect the study design. I suggest revising it to include the term "retrospective descriptive study" (example "Patient satisfaction after outpatient hysteroscopy: a retrospective descriptive study"), in accordance with reporting guidelines and to provide immediate context to readers.

Minor improvements to the English language are also recommended to enhance clarity and readability

Experimental design

The study presents original research within the scope of the journal. The research question is well defined and addresses a meaningful gap in current literature, namely, patient satisfaction after outpatient hysteroscopy.

The methodology is generally well described. The inclusion of all outpatient hysteroscopies during the specified period, regardless of indication, adds strength to the generalizability of findings. However, the satisfaction questionnaire used in the telephone survey should ideally be included as supplementary material to improve transparency and replicability.

The study meets ethical standards, and appropriate approvals and patient consent procedures are clearly stated.

Validity of the findings

The data are sound and statistically analyzed using basic but appropriate methods. The use of Spearman correlation is suitable for non-parametric associations. However, the findings would be further strengthened by a multivariable analysis (example: linear or logistic regression) to adjust for potential confounders such as age, menopausal status, or procedural indication.

The conclusions are supported by the data and aligned with the research question. The authors correctly note the influence of pre-procedure information and perceived pain on overall satisfaction.

Limitations are acknowledged, but the potential for selection bias (due to the exclusion of 285 out of 720 patients) and recall bias (from retrospective telephone surveys) should be more explicitly discussed.

Additionally, the authors should consider discussing how misinformation on social media may negatively impact patient perception and satisfaction with procedures such as hysteroscopy. Several recent studies have shown that online misinformation, particularly in social networks and forums, can foster fear, unrealistic expectations, or scepticism, thereby influencing reported outcomes. This is especially relevant in outpatient settings where procedures may already be perceived as uncomfortable or anxiety-inducing (PMID: 39118465 ; PMID: 38201027 ; PMID: 39729920)

---

## Round 0.2 · accepted · Accept

· Academic Editor

Accept

Thank you for revising your manuscript to address the concerns of the reviewers, both of whom now recommend acceptance. The manuscript is now ready for publication.

·

Basic reporting

No comment

Experimental design

No comment

Validity of the findings

No comment

Additional comments

I would like to sincerely thank the authors for carefully addressing the comments and suggestions provided during the review process. The revised manuscript shows clear improvements, particularly in the methodology and discussion sections, which now provide sufficient detail and a stronger contextualization within the existing literature.

The article is now clearer, more rigorous, and highlights its contribution to advancing patient-centred care in outpatient hysteroscopy. I believe the current version meets the standards required for publication, and I congratulate the authors for their thorough work and valuable contribution to the field.

Reviewer 2 ·

Basic reporting

The authors have incorporated all the suggestions into the manuscript. The article is suitable for publication

Experimental design

The authors have incorporated all the suggestions into the manuscript. The article is suitable for publication

Validity of the findings

The authors have incorporated all the suggestions into the manuscript. The article is suitable for publication